

# Construction of a predictive model for bone metastasis from first primary lung adenocarcinoma within 3 cm based on machine learning algorithm: a retrospective study

Yu Zhang[1], Lixia Xiao[2], Lan LYu[3] and Liwei Zhang[1]

[1] Department of Thoracic Surgery, First Affiliated Hospital of Xinjiang Medical University, Urumqi, Xinjiang, China
[2] Department of Thoracic Surgery, Feicheng Hospital Affiliated to Shandong First Medical University, Taian, Shandong, China
[3] Department of Plastic Surgery, Feicheng Hospital Affiliated to Shandong First Medical University, Taian, Shandong, China

Corresponding author
Liwei Zhang, zhangliweixj@163.com

## ABSTRACT

**Background**. Adenocarcinoma, the most prevalent histological subtype of non-small cell lung cancer, is associated with a significantly higher likelihood of bone metastasis compared to other subtypes. The presence of bone metastasis has a profound adverse impact on patient prognosis. However, to date, there is a lack of accurate bone metastasis prediction models. As a result, this study aims to employ machine learning algorithms for predicting the risk of bone metastasis in patients.

**Method**. We collected a dataset comprising 19,454 cases of solitary, primary lung adenocarcinoma with pulmonary nodules measuring less than 3 cm. These cases were diagnosed between 2010 and 2015 and were sourced from the Surveillance, Epidemiology, and End Results (SEER) database. Utilizing clinical feature indicators, we developed predictive models using seven machine learning algorithms, namely extreme gradient boosting (XGBoost), logistic regression (LR), light gradient boosting machine (LightGBM), Adaptive Boosting (AdaBoost), Gaussian Naive Bayes (GNB), multilayer perceptron (MLP) and support vector machine (SVM).

**Results**. The results demonstrated that XGBoost exhibited superior performance among the four algorithms (training set: AUC: 0.913; test set: AUC: 0.853). Furthermore, for convenient application, we created an online scoring system accessible at the following URL: https://www.xsmartanalysis.com/model/predict/?mid=731&symbol=7Fr16wX56AR9Mk233917, which is based on the highest performing model.

**Conclusion**. XGBoost proves to be an effective algorithm for predicting the occurrence of bone metastasis in patients with solitary, primary lung adenocarcinoma featuring pulmonary nodules below 3 cm in size. Moreover, its robust clinical applicability enhances its potential utility.

## INTRODUCTION

Lung cancer is recognized as one of the most prevalent and deadly malignancies worldwide (*Sung et al., 2021*). Epidemiological data reveals a morbidity rate of approximately 53.6 per 100,000 individuals, with an alarmingly high mortality rate of 45.6 per 100,000 individuals (*Siegel et al., 2022*). Notably, adenocarcinoma represents the predominant pathological subtype. The employment of low-dose spiral CT for lung cancer screening has resulted in the identification of an increasing number of lung cancers characterized by solitary nodules equal to or smaller than three cm (*Lancaster et al., 2021*; *Mazzone & Lam, 2022*; *Yang et al., 2022b*). While the prognosis for such lung cancers is often favorable due to a high rate of surgical resection, the onset of distant metastasis drastically diminishes the overall prognosis for the majority of patients.

The brain, bone, liver, and adrenal glands are recognized as the most prevalent sites of metastasis in lung cancer cases (*Buergy et al., 2021*; *Ferlay et al., 2018*; *Ma et al., 2022*). Notably, approximately one-third of lung cancer patients will experience bone metastasis (*Ferlay et al., 2018*; *Mazzone & Lam, 2022*; *Siegel et al., 2022*; *Zhang et al., 2019*), with adenocarcinoma being identified as a risk factor for this occurrence (*Chai et al., 2021*; *Cho et al., 2019*; *Hu et al., 2022*; *Huang et al., 2021*; *Liu et al., 2017*; *Wang et al., 2020*; *Zhang et al., 2019*). The presence of bone metastasis in lung cancer patients often signifies an unfavorable prognosis, with a median survival time of less than 1 year (*Hernandez et al., 2018*; *Hong et al., 2020*). Additionally, approximately half of patients with bone metastasis will develop skeletal-related events (SREs), such as pathological fractures, spinal cord compression, and hypercalcemia (*Siegel et al., 2022*). The manifestation of these events will bring significant physical, psychological, and economic burdens to patients, arising from both the associated ailments and the requisite radiotherapy and surgical interventions (*Cadieux et al., 2022*; *Decroisette et al., 2011*). Therefore, early detection and intervention are crucial in enhancing patient prognosis (*Yang et al., 2022a*; *Zhu et al., 2021*). However, the lack of specific symptoms prior to the occurrence of SREs, coupled with the limitations of current auxiliary examinations, including the low sensitivity of X-ray and CT scans, the low specificity of bone scans, as well as the high costs and low compliance associated with MRI and PET-CT scans (*Cook & Goh, 2020*; *Schmidkonz et al., 2019*; *Tal et al., 2021*; *Wood et al., 2018*; *Zhou et al., 2019*), underscores the necessity of identifying a sensitive and cost-effective method for predicting bone metastasis risk in different patient populations. Such an approach would facilitate the implementation of tailored follow-up strategies.

The Surveillance, Epidemiology, and End Results (SEER) database, established by the National Cancer Institute, contains a comprehensive collection of clinical data on various common tumors. This valuable resource includes information on approximately 25% of cancer patients in the United States each year, providing detailed data such as gender, age, and TNM staging. Machine learning (ML) is an interdisciplinary field that aims to simulate or replicate human learning processes using computers. It has gained significant attention in recent years and has been explored in lung cancer research. ML has shown promise in applications such as lung cancer diagnosis, classification of pathological types and genotypes, prediction of lymph node metastasis, treatment response assessment, and

prognosis prediction (*Gould et al., 2021*; *Gu et al., 2019*; *Koike et al., 2020*; *Wiesweg et al., 2020*; *Yoo et al., 2021*; *Yu et al., 2020b*). However, despite extensive research in the field, no prior studies have been found that specifically utilize machine learning to construct a prognostic model for lung adenocarcinoma patients with bone metastasis.

Therefore, we constructed prediction models based on different algorithms to evaluate the occurrence of bone metastases in patients with single lung cancer less than three cm, and compared the diagnostic performance of each algorithm to obtain the best prediction model, in order to provide personalized diagnosis and treatment for different patients. decision-making and more rational use of public health resources.

## MATERIALS & METHODS

### Study population

We retrieved a cohort of 234,770 patients diagnosed with non-small cell lung cancer between 2010 and 2015 from the SEER database, taking into account the absence of recorded metastatic sites of interest prior to 2010. Inclusion criteria consisted of: (1) Lung adenocarcinoma confirmed through tumor composite morphological coding criteria. (2) Pathological diagnosis confirmation. (3) Availability of complete follow-up data. Exclusion criteria were as follows: (1) Prior occurrence of malignant tumors other than lung cancer. (2) Inadequate information regarding T stage, N stage, Grade, Race, marital status, tumor site, and laterality. (3) Bilateral simultaneous lesions or overlapping lesions. (4) Tumor diameter exceeding 3 cm. (5) Unknown status of bone metastasis. In addition, an external validation set consisting of 125 eligible lung adenocarcinoma patients who underwent surgical treatment at Feicheng People's Hospital from January 2014 to December 2016 was included. This external dataset was incorporated to further assess the generalizability and robustness of our findings.

The study protocol received approval from the Ethics Committee of Feicheng People's Hospital and the research was granted an exemption from obtaining informed consent (Approval No: 202200201).

### Variable selection

Based on the findings of prior research (*Niu et al., 2019*; *Zhang et al., 2019*; *Zhou et al., 2017b*) and established expertise in the field, we opted to include nine variables in our model: age, sex, race, grade, T stage, N stage, tumor size, tumor site, and marital status. To determine correlations among these variables, we conducted the Spearman's test. Additionally, univariate and multivariate logistic regressions were performed to identify independent factors associated with bone metastasis. Furthermore, we employed a combination of importance ranking from each model to further refine the selection of variables. This rigorous screening process led us to the final set of variables with significant predictive value for bone metastasis in our study cohort.

### Predictive model construction and evaluation

In the model development phase, we employed seven machine learning algorithms, namely extreme gradient boosting (XGBoost), logistic regression (LR), light gradient boosting

machine (LightGBM), Adaptive Boosting (AdaBoost), Gaussian Naive Bayes (GNB), multilayer perceptron (MLP), and support vector machine (SVM). To ensure optimal performance, we conducted grid search CV to select the optimal hyperparameters for the model for each algorithm. This involved iteratively adjusting the model parameters to find the best combination that maximizes predictive accuracy and minimizes overfitting.

To evaluate the predictive capabilities of the models, we performed 10-fold cross-validation on both the training and validation datasets. This technique divides the data into ten subsets, trains the model on nine of them, and evaluates its performance on the remaining subset. By repeating this process with different subsets, we obtain a robust assessment of the model's generalization ability.

The performance of the models was assessed using various evaluation metrics, including receiver operating characteristic (ROC) curves, area under the curve (AUC), sensitivity, specificity, accuracy, and precision. ROC curves provide a visual representation of the trade-off between the true positive rate (sensitivity) and the false positive rate (1-specificity) at different classification thresholds. The AUC represents the overall discriminative power of the model. Sensitivity, specificity, accuracy, and precision provide additional insights into the model's performance across different evaluation dimensions.

In addition, decision curve analysis (DCA) allows for the comparison of predictive performance and potential practical application of various models by considering the threshold selection of actual decision risks and predicted probabilities. A calibration curve is employed to assess the predictive ability of the models and the consistency with actual situations.

Based on the best-performing model constructed, we further created an online calculator for bone metastases of lung adenocarcinoma.

## Statistical analysis

All Statistical analyses were performed using R version 3.6.3 (*R Core Team, 2021*) and Python version 3.7. Logreg 6.2.0 of R was used for logistic regression, and xgboost 1.2.1 and sklearn 0.22.1 of Python were used to rank the importance of the indicators in each model, build each model and evaluate its performance. sklearn 0.22.1 of Python is used to randomly split the data, and the random seed number is 1. statsmodels 0.11.1 was used for baseline data analysis. The Te Mann–Whitney U test and the chi-square test were used to compare continuous and categorical variables, respectively. The SMOTE module in the imbalanced-learn library of Python is used for sample balancing. The SciPy library can be used to perform Spearman correlation analysis in Python.

## RESULTS

### Basic characteristics of the study population

A total of 19,454 patients with lung adenocarcinoma presenting with a first primary solitary nodule ≤3 cm in diameter were included in our study (Table 1). The study encompassed a cohort of 8,029 male patients and 11,425 female patients, resulting in a total of 19,454 participants. Among these, 4,648 patients were classified as Grade I, 8,698 as Grade II, 5,990 as Grade III, and 118 as Grade IV, according to the grading system used. In terms

**Table 1 Baseline data table of patients included in the SEER database.**

| Characteristics | NBM $n = 18,239$ | BM $n = 1,215$ | P value |
|---|---|---|---|
| Sex, n (%) | | | <0.001 |
| Female | 10,845 (55.7%) | 580 (3%) | |
| Male | 7,394 (38%) | 635 (3.3%) | |
| Grade, n (%) | | | <0.001 |
| Grade I | 4,549 (23.4%) | 99 (0.5%) | |
| Grade II | 8,243 (42.4%) | 455 (2.3%) | |
| Grade III | 5,339 (27.4%) | 651 (3.3%) | |
| Grade IV | 108 (0.6%) | 10 (0.1%) | |
| Tumor Site, mean ± sd | 2.7671 ± 1.3715 | 2.7877 ± 1.425 | 0.615 |
| T Stage, n (%) | | | <0.001 |
| T1 | 12,476 (64.1%) | 557 (2.9%) | |
| T2 | 3,107 (16%) | 143 (0.7%) | |
| T3 | 1,587 (8.2%) | 218 (1.1%) | |
| T4 | 1,069 (5.5%) | 297 (1.5%) | |
| N Stage, n (%) | | | <0.001 |
| N0 | 13,687 (70.4%) | 312 (1.6%) | |
| N1 | 1,403 (7.2%) | 98 (0.5%) | |
| N2 | 2,529 (13%) | 557 (2.9%) | |
| N3 | 620 (3.2%) | 248 (1.3%) | |
| Tumor Size, mean ± sd | 19.789 ± 6.3503 | 22.573 ± 5.9824 | <0.001 |
| Race, n (%) | | | 0.006 |
| White | 16,599 (85.3%) | 1,077 (5.5%) | |
| Non-White | 1,640 (8.4%) | 138 (0.7%) | |
| Age, mean ± sd | 67.61 ± 10.149 | 66.337 ± 10.96 | <0.001 |
| Marital, n (%) | | | 0.043 |
| Yes | 10,205 (52.5%) | 716 (3.7%) | |
| No | 8,034 (41.3%) | 499 (2.6%) | |

**Notes.**

NMB, no bone metastasis; BM, bone metastasis.

of tumor size, 1,570 patients had tumors measuring less than 10 mm, 8,849 patients had tumors ranging from 11 to 20 mm, and 9,035 patients had tumors ranging from 21 to 30 mm. The patient selection process is shown in Fig. 1.

Considering the significant disparity between the number of patients with bone metastasis and those without, we employed the Synthetic Minority Over-sampling Technique (SMOTE) approach to balance the data, resulting in a proportion of 18,239:3,645 for bone metastasis to non-bone metastasis cases. The balanced dataset obtained through the oversampling method was subsequently divided into a training set and a test set, using a ratio of 7:3. The basic characteristics of the two datasets are presented in Table 2.

## Filtering of variables

The results of Spearman's test revealed that no significant correlation existed between the variables, as illustrated in Fig. 2. Furthermore, the univariate logistic regression analysis

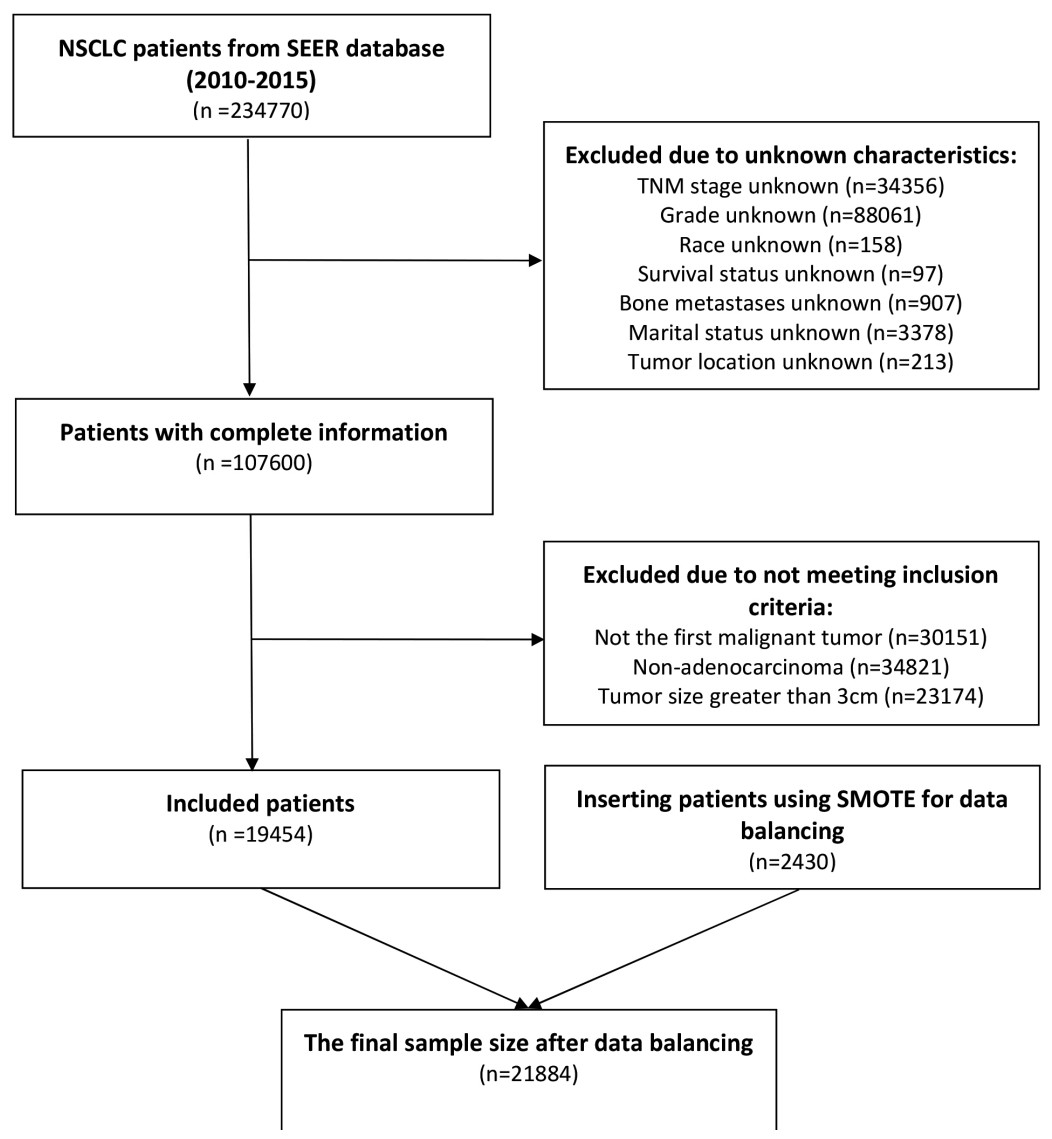

**Figure 1** SEER database patient selection process.

indicated that there was no significant association between tumor site and the occurrence of bone metastasis ($p = 0.615$).

However, in the multivariate logistic regression analysis, when considering other variables, such as tumor size, T stage, N stage, grade, and sex, all of them were found to have a significant association with the occurrence of bone metastases, suggesting their potential usefulness as predictors (Table 3).

In addition, we conducted an analysis to evaluate the importance of each variable in our machine learning algorithms. The results, as presented in Fig. 3, indicate that while there are minor differences in the importance rankings and proportions of variables across the algorithms, certain variables consistently ranked highly. Specifically, T stage, N stage,
**Table 2** Baseline data table of train and test tets after sample balancing.

| Characteristics | Train set | | Pvalue | Test set | | P value |
|---|---|---|---|---|---|---|
| | NBM (n = 12,762) | BM (n = 2,556) | | NBM (n = 5,477) | BM (n = 1,089) | |
| Sex, n (%) | | | 0.135 | | | 0.776 |
| Male | 5,206 (34%) | 1,002 (6.5%) | | 2,188(33.3%) | 430 (6.5%) | |
| Femal | 7,556 (49.3%) | 1,554 (10.1%) | | 3,289 (50.1%) | 659 (10%) | |
| Grade, n (%) | | | <0.001 | | | <0.001 |
| 1 | 3,172 (20.7%) | 255 (1.7%) | | 1,377 (21%) | 103 (1.6%) | |
| 2 | 5,766 (37.6%) | 1,237 (8.1%) | | 2,477 (37.7%) | 536 (8.2%) | |
| 3 | 3,747 (24.5%) | 1,058 (6.9%) | | 1,592 (24.2%) | 446 (6.8%) | |
| 4 | 77 (0.5%) | 6 (0%) | | 31 (0.5%) | 4 (0.1%) | |
| TumorSite, mean ± sd | 2.767 ± 1.3611 | 2.6099 ± 1.3242 | <0.001 | 2.7674 ± 1.3956 | 2.5849 ± 1.3041 | <0.001 |
| TStage, n (%) | | | <0.001 | | | <0.001 |
| T1 | 8,740 (57.1%) | 1,294 (8.4%) | | 3,736 (56.9%) | 545 (8.3%) | |
| T2 | 2,152 (14%) | 327 (2.1%) | | 955 (14.5%) | 136 (2.1%) | |
| T3 | 1,136 (7.4%) | 497 (3.2%) | | 451 (6.9%) | 229 (3.5%) | |
| T4 | 734 (4.8%) | 438 (2.9%) | | 335 (5.1%) | 179 (2.7%) | |
| NStage, n (%) | | | <0.001 | | | <0.001 |
| N0 | 9,524 (62.2%) | 696 (4.5%) | | 4,163 (63.4%) | 292 (4.4%) | |
| N1 | 1,008 (6.6%) | 315 (2.1%) | | 395 (6%) | 126 (1.9%) | |
| N2 | 1,807 (11.8%) | 1,247 (8.1%) | | 722 (11%) | 552 (8.4%) | |
| N3 | 423 (2.8%) | 298 (1.9%) | | 197 (3%) | 119 (1.8%) | |
| TumorSize, mean ± sd | 19.859 ± 6.3537 | 22.569 ± 5.854 | <0.001 | 19.625 ± 6.3399 | 22.017 ± 5.8392 | <0.001 |
| Race, n (%) | | | <0.001 | | | <0.001 |
| White | 11,616 (75.8%) | 2,308 (15.1%) | | 4,983 (75.9%) | 994 (15.1%) | |
| Black | 0 (0%) | 98 (0.6%) | | 0 (0%) | 33 (0.5%) | |
| Other | 1,146 (7.5%) | 150 (1%) | | 494 (7.5%) | 62 (0.9%) | |
| Age, mean ± sd | 67.562 ± 10.143 | 66.028 ± 10.838 | <0.001 | 67.721 ± 10.165 | 66.185 ± 10.658 | <0.001 |
| Marital, n (%) | | | <0.001 | | | <0.001 |
| Yes | 7,112 (46.4%) | 1,855 (12.1%) | | 3,093 (47.1%) | 777 (11.8%) | |
| No | 5,650 (36.9%) | 701 (4.6%) | | 2,384 (36.3%) | 312 (4.8%) | |

**Notes.**

NMB, no bone metastasis; BM, bone metastasis.

tumor size, grade, age, and sex consistently emerged as top-ranking variables in each algorithm. As a result, we selected these six variables as the final predictors to be included in our predictive model.

## Model performance and parameters

We compared the performance of models built by seven different algorithms in the training set and validation set, and the best performer was XGBoost Classifier (training set: AUC: 0.913; validation set: 0.860) (Figs. 4A and 4B). The performance of XGBoost model in the test set is basically consistent with the training set (AUC: 0.853) (Fig. 4C), which proves the good generalization ability of the model. The performance of each model in the training set and validation set is shown in Table 4.
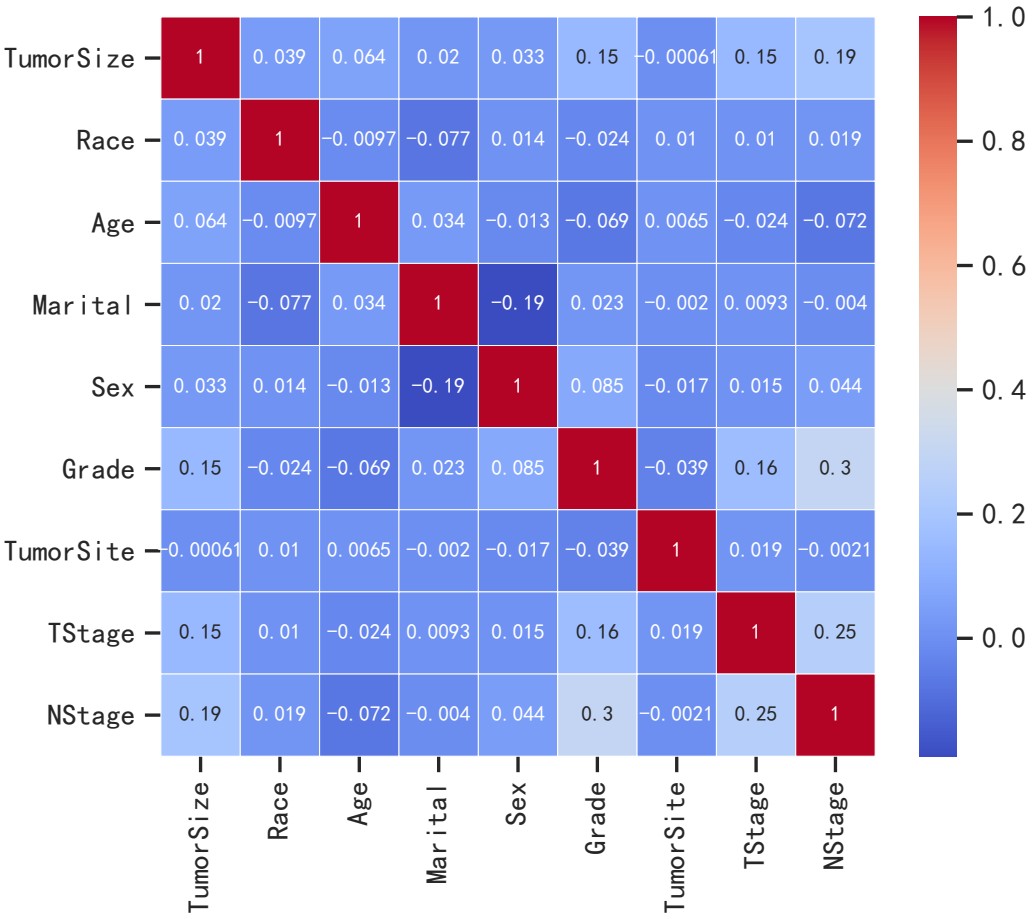

**Figure 2  Heatmap of Spearman correlation analysis for various variables.**

Furthermore, XGBoost demonstrated good performance in the external test set (AUC: 0.809) (Fig. 4D). The confusion matrices of the XGBoost model in the internal test set, and external test set also indicated high accuracy (Table 5). DCA analysis revealed that among all the models, the XGBoost model achieved the best decision effect (Fig. 4E). The calibration curve of XGBoost displayed the closest proximity to the diagonal line, indicating its superior reliability and stability (Fig. 4F).

## Online calculator

We built an online calculator based on XGBoost classifier model to assess a patient's risk of developing bone metastases. The calculator is accessible through the following URL: https://www.xsmartanalysis.com/model/predict?mid=731&symbol=7Fr16wX56AR9Mk233917 (Fig. 5). This user-friendly interface serves as a platform for healthcare professionals to input relevant patient data, which is then processed by the XGBoost classifier model to generate personalized risk predictions for bone metastasis occurrence.

**Table 3** Univariate and multivariate logistic regression analysis of variables.

| Characteristics | Total(N) | Univariate analysis | | Multivariate analysis | |
|---|---|---|---|---|---|
| | | Odds Ratio (95% CI) | *P* value | Odds Ratio (95% CI) | *P* value |
| Sex | 19,454 | | | | |
| Female | 11,425 | Reference | | Reference | |
| male | 8,029 | 1.606 (1.429–1.804) | <0.001 | 1.442 (1.271–1.637) | <0.001 |
| Grade | 19,454 | | | | |
| II | 8,698 | Reference | | Reference | |
| I | 4,648 | 0.394 (0.316–0.491) | <0.001 | 0.613 (0.487–0.771) | <0.001 |
| III | 5,990 | 2.209 (1.950–2.502) | <0.001 | 1.355 (1.184–1.551) | <0.001 |
| IV | 118 | 1.677 (0.872–3.228) | 0.121 | 0.994 (0.498–1.981) | 0.986 |
| Tumor Site | 19,454 | 1.011 (0.969–1.054) | 0.615 | | |
| T Stage | 19,454 | | | | |
| T1 | 13,033 | Reference | | Reference | |
| T3 | 1,805 | 3.077 (2.609–3.629) | <0.001 | 1.753 (1.468–2.092) | <0.001 |
| T2 | 3,250 | 1.031 (0.854–1.244) | 0.751 | 0.790 (0.650–0.961) | 0.018 |
| T4 | 1,366 | 6.223 (5.335–7.259) | <0.001 | 2.640 (2.228–3.128) | <0.001 |
| N Stage | 19,454 | | | | |
| N0 | 13,999 | Reference | | Reference | |
| N1 | 868 | 17.547 (14.582–21.116) | <0.001 | 9.749 (7.986–11.900) | <0.001 |
| N2 | 1,501 | 3.064 (2.426–3.870) | <0.001 | 2.236 (1.760–2.840) | <0.001 |
| N3 | 3,086 | 9.662 (8.358–11.169) | <0.001 | 6.107 (5.228–7.135) | <0.001 |
| Tumor Size | 19,454 | 1.077 (1.066–1.088) | <0.001 | 1.044 (1.033–1.055) | <0.001 |
| Race | 19,454 | | | | |
| White | 17,676 | Reference | | Reference | |
| Non-White | 1,778 | 1.297 (1.079–1.559) | 0.006 | 1.146 (0.937–1.401) | 0.184 |
| Age | 19,454 | 0.988 (0.982–0.994) | <0.001 | 0.997 (0.991–1.003) | 0.276 |
| Marital | 19,454 | | | | |
| Yes | 10,921 | Reference | | Reference | |
| No | 8,533 | 0.885 (0.787–0.996) | 0.043 | 0.898 (0.789–1.021) | 0.100 |

**Notes.**
CI, Confidence Interval.

# DISCUSSION

Bone metastasis is a common occurrence in cases of lung cancer, with research studies reporting that approximately 10%–40% of lung cancer patients experience bone metastasis (*D'Oronzo, Wood & Brown, 2021*). However, in our study, which included 19,454 patients diagnosed with lung adenocarcinoma, only approximately 6.25% of the cases presented with bone metastasis. This proportion is lower than what has been previously reported. The discrepancy could possibly be attributed to the fact that our study solely relied on data collected by the Surveillance, Epidemiology, and End Results (SEER) program, which recorded bone metastasis occurrences at the time of data collection without comprehensive follow-up information. Furthermore, the majority of patients included in our study

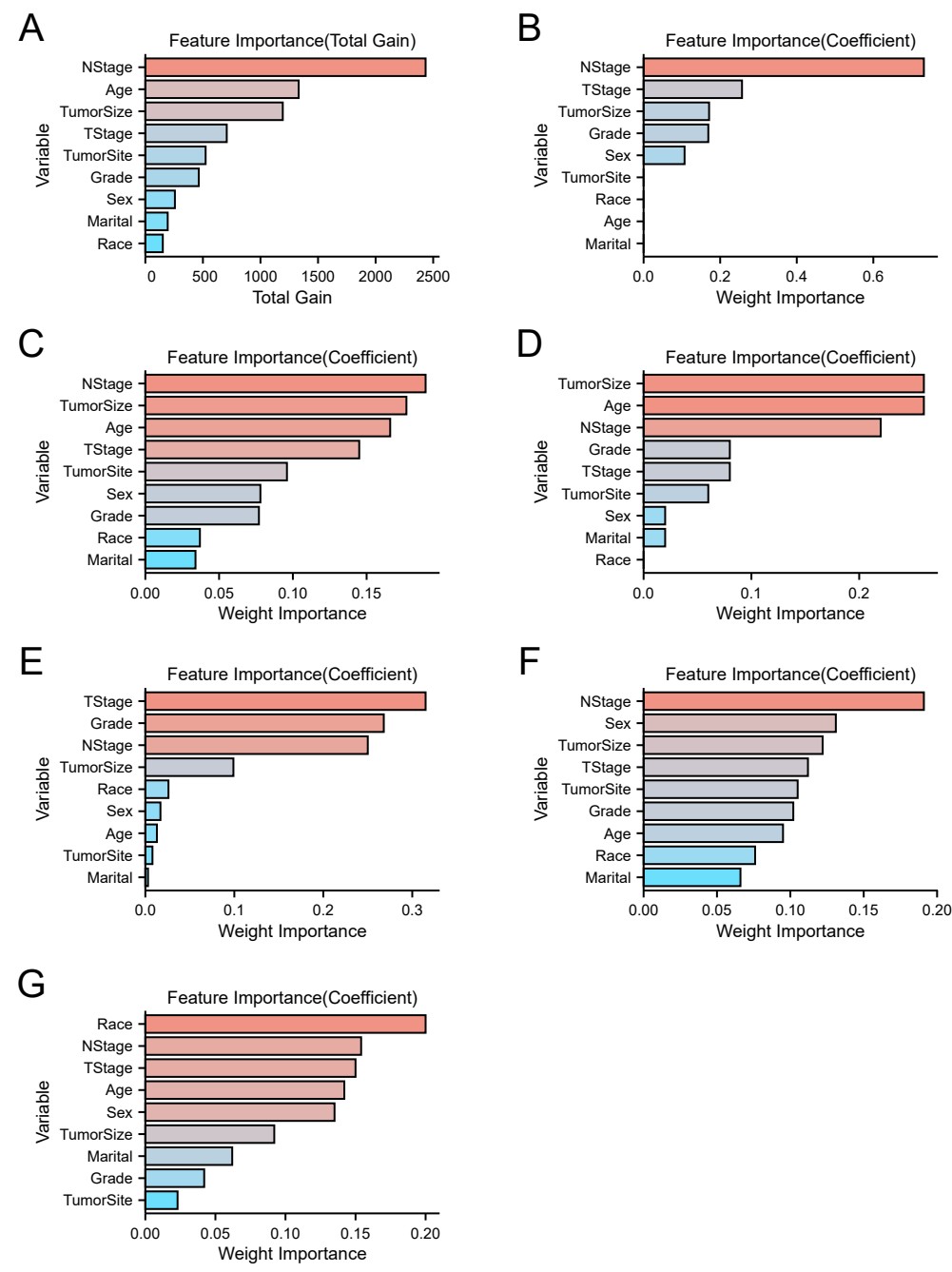

**Figure 3** Variable importance ranking plot based on various machine learning algorithms. (A) XG-Boost. (B) LR. (C) LightGBM. (D) AdaBoost. (E) GNB. (F) MLP. (G) SVM.

presented with smaller tumor diameters and earlier pathological stages, suggesting a lower probability of bone metastasis occurrence.

The current clinical diagnostic and therapeutic techniques for bone metastasis in lung cancer still fail to meet the needs of early detection and treatment. This is primarily due to the lack of specific clinical manifestations in the early stages, and the NCCN lung cancer
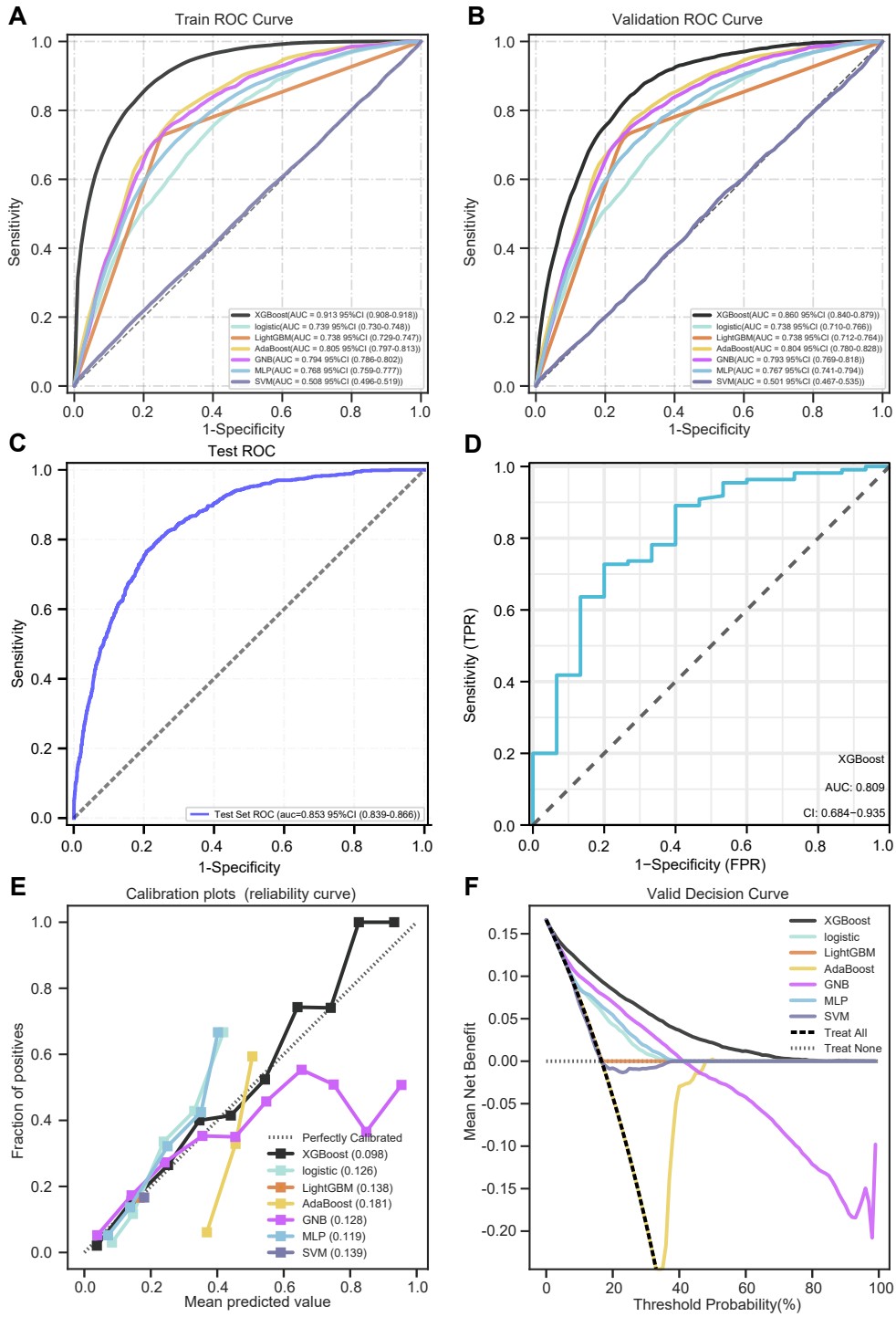

**Figure 4   ROC, DCA, and calibration curve of the prediction model.** (A) ROC curves of the seven models on the training set. (B) ROC curves of the seven models on the validation set. (C) ROC curves of XGBoost models on the internal test set. (D) ROC curves of XGBoost models on the external test set. (E) DCA curve of the seven models. (F) Calibration curves of the seven models.

**Table 4    Predictive performance of different models on training and validation sets.**

| Model | AUC (SD) | Cutoff (SD) | Accuracy (SD) | Sensitivity (SD) | Specificity (SD) | F1 score (SD) |
|---|---|---|---|---|---|---|
| Training set | | | | | | |
| XGBoost | 0.913(0.002) | 0.200(0.025) | 0.810(0.015) | 0.858(0.023) | 0.800(0.022) | 0.600(0.012) |
| LR | 0.739(0.001) | 0.162(0.002) | 0.625(0.009) | 0.757(0.013) | 0.599(0.013) | 0.402(0.001) |
| LightGBM | 0.738(0.001) | 0.167(0.000) | 0.834(0.000) | 0.726(0.003) | 0.750(0.001) | NaN |
| AdaBoost | 0.805(0.001) | 0.397(0.005) | 0.725(0.006) | 0.782(0.009) | 0.711(0.008) | 0.485(0.003) |
| GNB | 0.794(0.001) | 0.134(0.003) | 0.738(0.002) | 0.744(0.005) | 0.737(0.003) | 0.486(0.001) |
| MLP | 0.768(0.022) | 0.170(0.008) | 0.697(0.039) | 0.735(0.016) | 0.689(0.048) | 0.448(0.026) |
| SVM | 0.508(0.059) | 0.180(0.044) | 0.607(0.189) | 0.411(0.306) | 0.646(0.287) | 0.205(0.116) |
| Validation set | | | | | | |
| XGBoost | 0.860(0.008) | 0.200(0.025) | 0.781(0.021) | 0.837(0.046) | 0.742(0.041) | 0.556(0.026) |
| LR | 0.738(0.009) | 0.162(0.002) | 0.625(0.009) | 0.766(0.045) | 0.601(0.046) | 0.402(0.010) |
| LightGBM | 0.738(0.010) | 0.167(0.000) | 0.834(0.000) | 0.726(0.024) | 0.750(0.013) | NaN |
| AdaBoost | 0.804(0.010) | 0.397(0.005) | 0.723(0.012) | 0.764(0.050) | 0.736(0.044) | 0.481(0.017) |
| GNB | 0.793(0.010) | 0.134(0.003) | 0.738(0.010) | 0.748(0.037) | 0.740(0.025) | 0.487(0.014) |
| MLP | 0.767(0.020) | 0.170(0.008) | 0.696(0.039) | 0.736(0.043) | 0.698(0.041) | 0.449(0.034) |
| SVM | 0.501(0.057) | 0.180(0.044) | 0.603(0.191) | 0.486(0.297) | 0.578(0.287) | 0.217(0.094) |

Notes.
   XGBoost, extreme gradient boosting; LR, logistic regression; LightGBM, light gradient boosting machine; AdaBoost, Adaptive Boosting; GNB, Gaussian Naive Bayes; MLP, multilayer perceptron; SVM, support vector machine.

**Table 5    Confusion matrix of XGBoost model on internal and external test sets.**

| Internal Test Set | | Actual | | External Test Set | | Actual | |
|---|---|---|---|---|---|---|---|
| | | BM | NBM | | | BM | NBM |
| Predictive | BM | 778 | 973 | Predictive | BM | 7 | 5 |
| | NBM | 311 | 4504 | | NBM | 8 | 105 |

Notes.
   NMB, no bone metastasis; BM, bone metastasis.

treatment guidelines do not recommend routine skeletal imaging examinations to rule out bone metastasis in asymptomatic lung cancer patients. Therefore, clinicians typically only conduct relevant examinations for patients who exhibit obvious clinical symptoms such as bone pain, pathological fractures, spinal cord compression, and hypercalcemia. However, patients presenting with such symptoms often have already experienced skeletal-related events (SREs) and missed the optimal timing for early treatment (*Wood et al., 2018*). Research has reported that the majority of lung cancer patients with bone metastasis will experience SREs. Apart from causing pain, SREs lead to loss of physical function, significantly shorter survival, and adverse physiological and psychological health outcomes (*Anton et al., 2021*; *Brouns et al., 2021*; *Li et al., 2022a*; *Qin et al., 2021*; *Sethakorn et al., 2022*). To better identify patients at higher risk of developing bone metastasis and assist clinicians in formulating appropriate diagnostic, therapeutic, and follow-up plans, we validated several advanced machine learning algorithms to predict bone metastasis in adenocarcinoma patients with tumor size less than 3 cm.

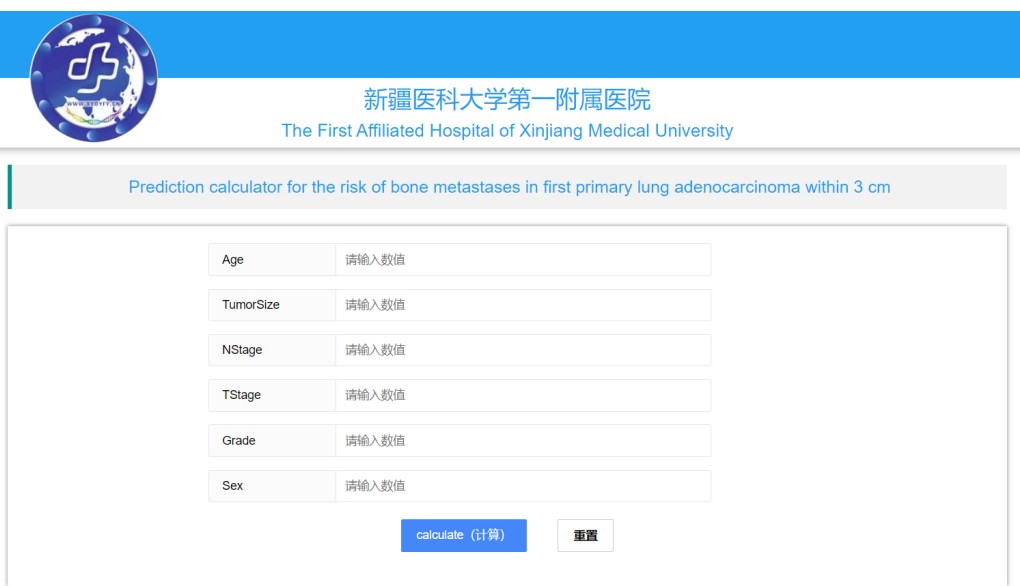

**Figure 5  Screenshot of the web page for the online rating system.**

In this study, we employed XGBoost, LR, LightGBM, AdaBoost, GNB, MLP and SVM algorithms for model construction and compared the diagnostic capabilities of different algorithms. Gradient boosting machine (GBM) serves as an upgraded machine learning technique, known for its ability to transform weak learners into strong learners, thereby enhancing model predictive performance. XGBoost, an enhanced version of GBM, has been particularly favored for its shorter computational time and higher accuracy (*Sheridan et al., 2016*), making it widely applied in disease prediction tasks encompassing diagnosis, survival, and prognosis (*Chen et al., 2019*; *Hou et al., 2020*; *Khera et al., 2021*; *Yu et al., 2020a*). Our results demonstrated that the XGBoost algorithm exhibited the best predictive performance. This model holds promise in aiding clinicians to predict the risk of bone metastasis in patients and encouraging further investigations for high-risk individuals to facilitate early detection and improve prognosis. Moreover, when selecting adjuvant therapies for high-risk bone metastasis patients, consideration should be given to the impact of treatment on bone tissue.

In this study, a comprehensive analysis of various machine learning algorithms revealed that the most influential predictors of bone metastasis (BM) include T stage, N stage, grade, sex, age, and tumor size. Notably, *Wang et al. (2017)* reported that adenocarcinoma and stage III pathological stages are associated with an increased risk of bone metastasis. This finding suggests that the incidence of bone metastasis is higher in patients with adenocarcinoma compared to other types of lung cancer, underscoring the clinical significance of our study. However, the underlying mechanisms responsible for the elevated occurrence of bone metastasis in patients with lung adenocarcinoma are presently poorly understood. Some studies have postulated that the upregulation of vascular endothelial growth factor (VEGF) in adenocarcinoma may play a crucial role in promoting bone

metastasis. VEGF is known to be a pivotal factor in tumor angiogenesis and is considered a prerequisite for tumor metastasis (*Muench et al., 2019*). Moreover, adenocarcinoma typically originate from mucous cells or goblet cells located at the periphery of lung tissue, rendering them prone to invading both blood and lymphatic vessels. Consequently, they exhibit a predilection for distant metastasis or local invasion, often involving neighboring ribs or the sternum (*Nagata et al., 2013*; *Wang et al., 2019*).

In the context of variable selection processes in machine learning research, it was observed that age exhibited no significant association with the outcome according to the multivariate Cox analysis. However, despite this finding, age consistently ranked highly in the variable importance rankings generated by multiple machine learning algorithms, indicating its potential predictive capability or informational value within the model. Furthermore, numerous studies have corroborated age as a significant risk factor for bone metastasis in patients with lung cancer. These findings collectively highlight the potential relevance of age as a predictive feature and emphasize its importance in evaluating the risk of bone metastasis in this patient population. The study conducted by *Da Silva, Bergmann & Thuler (2019)* confirmed a significant association between age and adenocarcinoma in relation to the occurrence of Bone Metastasis. *Zhou et al. (2017a)* revealed that age, concentrations of neuron-specific enolase, and histopathological types independently correlated with the incidence of bone metastases in patients with lung cancer.

The results of a systematic review showed that T4 and N3 are risk factors for bone metastasis in patients with lung cancer (*Niu et al., 2019*). Studies have shown that male lung cancer patients are more likely to develop bones (*Brouns et al., 2021*; *Qin et al., 2021*). Research by *Ma et al. (2019)* showed that although lung cancer is a non-sex-specific tumor, sex-related hormones may affect the occurrence of bone metastases. There are also studies that suggest that the high rate of bone metastasis in men may be related to the higher rate of smoking in men. *Li et al. (2022b)* showed that bone metastasis in NSCLC was associated with higher grade and later T stage. In our study, T stage, N stage, grade and sex are all independent risk factors for bone metastasis and have a higher proportion in the importance ranking of the ML algorithm, so that the results of previous related studies are consistent.

The univariate and multivariate analyses indicated that tumor site, race, and marital status were not significantly associated with bone metastasis, as they ranked lower in importance in most machine learning classification algorithms. Furthermore, previous studies (*Da Silva, Bergmann & Thuler, 2019*; *Hu et al., 2022*; *Niu et al., 2019*; *Zhou et al., 2017b*) have shown no significant correlation between race and marital status with bone metastasis, therefore they were excluded from consideration. A study conducted by *Hu et al. (2022)* suggested that tumor site could be a risk factor for bone metastasis. However, their study included all lung cancer patients with bone metastasis, without distinguishing histological types, which differs from our study population. Additionally, the study excluded all patients who did not develop bone metastasis during follow-up, which we believe might introduce bias. Therefore, tumor site was not included in our model.

There have been several studies on predictive models for bone metastasis in patients with lung cancer. *Li et al. (2022b)* conducted an analysis using the SEER database and developed

a predictive model for bone metastasis in non-small cell lung cancer patients using the XGBoost algorithm. The model demonstrated the best performance in both internal and external validation datasets, with AUC scores of 0.808 and 0.841, respectively. In another study, *Teng et al. (2020)* developed a diagnostic molecular model for bone metastasis using four bone biochemical markers (OPG, PTHrP, TPINP, β-CTX). The model achieved a sensitivity of 85.7% and specificity of 87.5%, and the average predictive time for bone metastasis occurrence was 9.46 months earlier than whole-body bone imaging. *Zhu et al. (2021)* established a multivariate regression model incorporating four bone metabolism markers (β-CTX, TPINP, calcium (Ca), phosphorus (P) by examining bone metabolism-related indicators in 339 patients with non-metastatic lung cancer, lung cancer with bone metastasis, and benign lung diseases. The model exhibited a sensitivity of 70.0%, specificity of 91.0%, a positive predictive value of 82.5%, and a negative predictive value of 72.0%. The study conducted by *Zhang et al. (2019)* developed a nomogram for predicting bone metastasis in lung adenocarcinoma, demonstrating high diagnostic performance (AUC: 0.83; 95% CI [0.796–0.809]). Our model specifically targets adenocarcinoma cases with a size below three cm, which represents the most prevalent type in current clinical practice. Through external validation using independent datasets, our model has demonstrated superior diagnostic accuracy and generalizability, thus enhancing its suitability for clinical applications.

The existing models have generally characterized the study population as all lung cancer or NSCLC patients, and many of these models have included variables that are not commonly utilized in clinical practice. In our study, the research population was defined as a specific group of adenocarcinoma patients with tumor sizes less than three cm. This selection was based on the consideration that the most common size range for lung cancer in clinical practice is within three cm, and adenocarcinoma is the most prevalent histological type. Additionally, a significant proportion of small lung cancers within this size range do not receive adjuvant therapy. Therefore, we deemed it necessary to focus on studying this specific population. However, due to the lower risk of bone metastasis in this category of lung cancer compared to larger tumor sizes, only approximately 6% of the patients included in our study presented bone metastasis. This resulted in a class imbalance issue during the model development process. To mitigate the bias and inaccuracy caused by the class imbalance and enhance the generalization ability of the model as well as the reliability of performance evaluation, we employed SMOTE to balance the samples. Although SMOTE, which is a widely used oversampling technique, interpolates between available minority class samples to generate additional data, there is a potential risk of introducing noise into the dataset through the synthesis of new samples. However, despite this concern, our model exhibits robust classification capabilities, as evidenced by its consistently strong performance on both internal and external test sets, even after applying SMOTE for resampling. This indicates the effectiveness and reliability of our model in accurately classifying the target variable.

In the internal test set, the model predicted 778 cases as BM, which were indeed BM (true positives), and it correctly identified 4,504 cases as NBM, which were actually NBM (true negatives) (Table 5). However, there were 973 cases that the model predicted as BM

which were actually NBM (false positives), and 311 cases were predicted as NBM but were actually BM (false negatives). In the external test set, the model predicted seven cases as BM, which were true positives and correctly identified 105 cases as NBM (true negatives). There were five false positives (predicted as BM but were actually NBM) and eight false negatives (predicted as NBM but were actually BM). The results indicate that the model has a higher number of true negatives and true positives compared to false negatives and false positives, suggesting a reasonable level of accuracy in prediction. The true negative rate is especially high, which is positive for a screening test where the aim is to minimize the number of cases that go undetected. However, the false positives in the internal test set are relatively high and could be a concern, potentially leading to unnecessary anxiety and additional testing for those patients. Moreover, when comparing the performance in the internal and external test sets, the model seems to maintain its predictive ability in an external population, although the sample size for the external test set is quite small, and this could affect the reliability of the generalization. To fully evaluate the model's performance, it would be important to calculate metrics such as sensitivity, specificity, positive predictive value, negative predictive value, and the area under the ROC curve. These metrics could provide more comprehensive insights into how well the model performs and how it might be improved.

In recent years, with the popularity of lung cancer screening, patients with lung adenocarcinoma with peripheral lung nodules have been increasing year by year. However, there are few studies on the risk of bone metastasis in this type of patients, and there is no research on applying the ML algorithm to the prediction of bone metastasis in patients with lung adenocarcinoma. To the best of our knowledge, our research is the first report on the application of ML algorithms to develop this type of model.

Our study has some limitations: (1) This study was a retrospective analysis, which may introduce bias. Therefore, we still need prospective clinical studies to further confirm our conclusions. (2) The use of SEER data lacks subsequent bone metastasis data, which prevents us from including patients with new bone metastases during the follow-up process. (3) The lack of clinical blood test data makes it impossible for us to use them as variables for importance evaluation and model construction. (4) The time to onset of bone metastases could not be analyzed because the time to onset of bone metastases was not recorded. (5) The SEER data may not mirror the specific population characteristics of Feicheng dataset (external validation), which could affect the external validation results, such as leading to potential biases, particularly in model performance, as a model trained on SEER data might not generalize well to the Feicheng dataset if these underlying differences are not accounted for. Such variability highlights the need for cautious interpretations of the predictive model's applicability and the importance of considering regional differences in clinical studies.

Despite the valuable insights gained from our study, it is important to acknowledge its limitations to ensure the validity and applicability of our findings. Firstly, it is crucial to note that our study design was retrospective in nature, which may introduce inherent biases. Thus, further confirmation of our conclusions is warranted through prospective clinical studies. Secondly, the use of the SEER database as our data source has its limitations.

A significant drawback is the absence of subsequent bone metastasis data, hindering the inclusion of patients who developed new bone metastases during the follow-up period. This absence may result in an incomplete representation of the true incidence of bone metastasis in our study population. Furthermore, the lack of available clinical blood test data restricts our ability to incorporate these variables into our models for importance evaluation and model construction. This limitation may have an impact on the overall accuracy and comprehensiveness of our prediction models. Lastly, due to the unavailability of recorded data on the time to onset of bone metastases, we were unable to analyze and incorporate this parameter into our study. This absence limits our ability to assess the timing and progression of bone metastasis development.

To address these limitations, future research should consider prospective study designs, inclusion of comprehensive clinical data, and meticulous recording of relevant variables such as time to onset of bone metastases. By addressing these limitations, we can enhance the robustness and applicability of our findings, thereby facilitating more accurate and reliable personalized diagnosis and treatment decision-making for lung adenocarcinoma patients with potential bone metastasis.

## CONCLUSIONS

In summary, we developed a predictive model for bone metastasis in patients with a single lung adenocarcinoma using the XGBoost algorithm. The model considers age, T stage, N stage, grade, sex, and tumor size as characteristic variables. Our evaluation demonstrated excellent diagnostic capabilities, indicating the model's potential for guiding diagnosis and treatment strategies in clinical practice. However, further validation and refinement are necessary, and additional clinical variables could enhance its accuracy and utility. Nevertheless, our model offers valuable insights for personalized decision-making in managing lung adenocarcinoma patients at risk of bone metastasis.

## ACKNOWLEDGEMENTS

We would like to acknowledge the support of Extreme Smart Analysis platform for their contribution to the architecture of online calculator.

### Funding
The authors received no funding for this work.

### Competing Interests
The authors declare there are no competing interests.

### Author Contributions
- Yu Zhang conceived and designed the experiments, performed the experiments, analyzed the data, prepared figures and/or tables, and approved the final draft.

- Lixia Xiao performed the experiments, analyzed the data, prepared figures and/or tables, and approved the final draft.
- Lan LYu performed the experiments, analyzed the data, prepared figures and/or tables, and approved the final draft.
- Liwei Zhang conceived and designed the experiments, performed the experiments, analyzed the data, prepared figures and/or tables, authored or reviewed drafts of the article, and approved the final draft.

## Human Ethics

The following information was supplied relating to ethical approvals (*i.e.*, approving body and any reference numbers):

The Ethics Committee of Feicheng People's Hospital.

## Data Availability

The SEER data is available at figshare: Zhang, Yu (2023). 2010-2015 SEER-LUAD.zip. figshare. Dataset. https://doi.org/10.6084/m9.figshare.24481693.v1.

The patient information from Feicheng City People's Hospital is available in the Supplemental File.

## Supplemental Information

Supplemental information for this article can be found online at http://dx.doi.org/10.7717/peerj.17098#supplemental-information.

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
