# Peer review of "Construction of a predictive model for bone metastasis from first primary lung adenocarcinoma within 3 cm based on machine learning algorithm: a retrospective study"

_PeerJ, doi:10.7717/peerj.17098_

## Round 0.1 · original submission · Major Revisions

I have the following two comments on the article for the authors as follows:

(1) Please pay more attention to detail when the authors revise the article. The link mentioned on line 38 seems not work is just because the authors mistakenly included “]” into the link from my point of view. So please avoid such errors in the revised version.

(2) The figures and tables were not prepared well. A high-quality paper always needs high-quality pictures and tables to back it up. Figure 1 is too simple to be beautiful. The authors need to draw the selection flowchart hierarchically according to inclusion and exclusion criteria. For this point, the authors can refer to many meta-analysis articles. Figure 3 is not clear enough and is also overall-uncoordinated. Figure 4 and Figure 5 can be combined into a compact good picture as long as the authors can master certain skills. Finally, why do the authors use the single line spacing in the main body of the text but use the double line spacing in the table? Usually, the table needs to be made more compact, and a complete table should be displayed or printed on one page.

When revising your manuscript, please consider all issues mentioned in the comments from the two reviewers and the Editor carefully and provide suitable responses for any comments. Please note that your revised submission may need to be re-reviewed.

PeerJ values your contribution and I look forward to receiving your revised manuscript.

Reviewer 1 ·

Basic reporting

The article meets the journal’s guidelines. Ethical approval statements have been checked.
The data has been deidentified and the experiments conducted have been ethical.
The supplemental files and the figures and tables in the manuscript have been checked.
Overall, I commend the authors that the research layout is clear to understand. Professional English language has been used throughout. Figures are relevant and labelled. However, major revisions are required.

Experimental design

no comment

Validity of the findings

The xsmart link mentioned on line 38 is not accessible and shows error 404 (maybe because I am trying to access it from the US?). Check Python spelling on line 149. Report the exact p-value on line 174 instead of > 0.05. It is not clearly mentioned which grid search technique for hyperparameter tuning is used for the ML models (randomized grid search cv or grid search cv). Code needs to be made available on Github or any other platform. Unavailability of the code makes the analysis weaker to interpret and impossible to reproduce. Some terms are used interchangeably in the manuscript: univariable and univariate, multivariable and multivariate are. Please keep consistent use of terminologies.

Reviewer 2 ·

Basic reporting

This study conducted constructed prediction models based on different algorithms to evaluate the occurrence of bone metastases in patients with single lung cancer less than 3 cm, and compared the diagnostic performance of each algorithm to obtain the best prediction model. However, there are some key issues that need to be addressed. In my opinion, a major revision is required before further publication.

Experimental design

1. The data on comorbidities and anti-cancer treatment is not included in your database. However, they may be important factors in bone metastasis. Please explain the reasons for excluding these factors.
2. You applied SEER database to construct predictive model, then used patient data in Feicheng Hospital for external validation. Lastly, the related software was used in Urumqi. However, the demographics of patients in these three areas of study was quite different.
3. How do you deal with the missing data?

Validity of the findings

4. In Table 1-3, appears to be a potential factor for bone metastasis. Please discuss it.
5. In Table 1, race was divided white and non-white. In Table 2, race was divided into 1,2,3. What are the race 1-3 and marital 1-2?
6. As shown in Table 5, the model has a high truth-false ratio. Please explain the reasons.

Additional comments

7. The previous studies have applied the nomogram to predict bone metastasis in patients with lung adenocarcinoma. Please compare your results with these studies.

---

## Round 0.2 · Minor Revisions

When revising your manuscript, please consider all issues mentioned in the comments from the two reviewers and provide suitable responses for any comments. In addition, please proofread your manuscript (including literature cited) and supplementary files carefully before your submission.

Reviewer 2 ·

Basic reporting

no comment

Experimental design

no comment

Validity of the findings

1. The Baseline demographic comparison among SEER patients, Feicheng patients, and Urumuqi patients should be conducted.

2. Why do you think "the missing values were determined to be due to random factors"? How many cases did you remove?

3. In Table 3, sex was divided into 1,2. What are the sex 1-2?

4. In Table 5, the predictive model predicted there were 778+973 BM cases. However, only 778 were actual BM cases. Please discuss the high false positive rate.

5. The heterogeneity of internal and external samples should be discussed in the Limitation part.

Additional comments

no comment

·

Basic reporting

In this manuscript, the authors have developed a predictive model for anticipating the risk of bone metastasis in patients with lung adenocarcinoma and primary tumors smaller than 3 cm. The evaluation involves a comparative analysis of various machine learning algorithms' diagnostic efficacy. In light of the article's content, I have the major concerns listed below:

1. The numerical order of N stage in Tables 1-3 requires correction. The N stage in the TNM (Tumor, Node, Metastasis) staging system is used to describe the extent of lymph node involvement, ranging from N0 to N3 without N4. Simultaneously, the N1, N2, N3 and N4 used in the charts should be respectively corrected to N0, N1, N2 and N3.

2. Based on the data utilized by the model, you have considered both early stage cases (T1 and T2) and mid-to-late stage cases (T3, T4, as well as N2, N3), and the occurrence of bone metastasis is elevated in mid-to-late stage compared to the early stage.Would the inclusion of mid-to-late-stage data affect the accuracy of the model in assessing early-stage cases?

Experimental design

no comment

Validity of the findings

no comment

---

## Round 0.3 · Minor Revisions

Before resubmission, please carefully review the entire manuscript for errors. For examples, in Figure 1, included patients (n=19454) does not equal to the number (n=19914) that Patients with complete information (n=107600) minus the exluded ones (29691+34821+23174). In Table 2, the BM number of Test set should be 1089 rather than 12762.

---

## Round 0.4 · accepted · Accept

The authors have responded well to the comments of the three reviewers.